# Effects of Para-Toluenesulfonamide on Canine Melanoma Xenotransplants in a BALB/c Nude Mouse Model

**DOI:** 10.3390/ani12172272

**Published:** 2022-09-02

**Authors:** Chien-Teng Lin, Chuen-Fu Lin, Jui-Te Wu, Hsiao-Pei Tsai, Shu-Ying Cheng, Huei-Jyuan Liao, Tzu-Chun Lin, Chao-Hsuan Wu, Yu-Chin Lin, Jiann-Hsiung Wang, Geng-Ruei Chang

**Affiliations:** 1Ph.D. Program of Agriculture Science, National Chiayi University, 300 University Road, Chiayi 60004, Taiwan; 2Department of Veterinary Medicine, National Chiayi University, 580 Xinmin Road, Chiayi 60054, Taiwan; 3Department of Veterinary Medicine, College of Veterinary Medicine, National Pingtung University of Science and Technology, 1 Shuefu Road, Neipu, Pingtung 912301, Taiwan; 4Department of Pet Medicine, Gongwin Biopharma Co., Ltd., 1 Section, 80 Jianguo North Road, Zhongshan District, Taipei 104001, Taiwan

**Keywords:** apoptosis, canine melanoma, cisplatin, inflammation, metastasis, para-toluenesulfonamide

## Abstract

**Simple Summary:**

Canine melanomas are malignant neoplasms, and primary melanomas arise at the footpad, nail bed, gastrointestinal tract, mucocutaneous junction, and eyes. Para-toluenesulfonamide (PTS) is a small molecule that acts against several cancers (hepatocellular carcinoma, nonsmall-cell lung cancer, and tongue squamous cell carcinoma). The pharmacological pathway of PTS has the potential to exert anti-inflammatory and antithrombotic functions. We established canine melanoma xenografts in mice and randomized the animals into four treatment groups: saline, cisplatin, PTS, and PTS combined with cisplatin. The findings indicated that compared with the control mice, mice treated with PTS and the combination of PTS and cisplatin showed retarded tumor growth; increased tumor apoptosis through the upregulation of caspase 3 and extracellular signal-regulated kinase phosphorylation; decreased inflammation levels of cytokines, such as interleukin-1β, tumor necrosis factor-α, and interleukin-6; reduced inflammation-related factors, such as the cyclooxygenase-2 protein and nuclear factor-κB mRNA; enhanced anti-inflammation-related factors; and inhibition of the metastasis-related factors transforming growth factor β, CD44, epidermal growth factor receptor, and vascular endothelial growth factor. Combining cisplatin with PTS has a stronger effect than PTS alone. These findings may prove useful in further explorations of the application of PTS or PTS combined with cisplatin to the treatment of canine melanoma in general.

**Abstract:**

The pharmacological pathway of para-toluenesulfonamide (PTS) restricts the kinase activity of the mammalian target of rapamycin, potentially leading to reductions in cell division, cell growth, cell proliferation, and inflammation. These pathways have a critical effect on tumorigenesis. We aimed to examine the antitumor effect of PTS or PTS combined with cisplatin on canine melanoma implanted in BALB/c nude mice by estimating tumor growth, apoptosis expression, inflammation, and metastasis. The mice were randomly divided into four groups: control, cisplatin, PTS, and PTS combined with cisplatin. Mice treated with PTS or PTS combined with cisplatin had retarded tumor growth and increased tumor apoptosis through the enhanced expression of cleaved caspase 3 and extracellular signal-regulated kinase phosphorylation, decreased inflammatory cytokine levels, reduced inflammation-related factors, enhanced anti-inflammation-related factors, and inhibition of metastasis-related factors. Mice treated with PTS combined with cisplatin exhibited significantly retarded tumor growth, reduced tumor size, and increased tumor inhibition compared with those treated with cisplatin or PTS alone. PTS or PTS combined with cisplatin could retard canine melanoma growth and inhibit tumorigenesis. PTS and cisplatin were found to have an obvious synergistic tumor-inhibiting effect on canine melanoma. PTS alone and PTS combined with cisplatin may be antitumor agents for canine melanoma treatment.

## 1. Introduction

Melanoma, which develops from melanocytes, is a commonly occurring cancer in dogs [1]. These tumors exhibit an extremely variable biological behavior and are best characterized based on their site, size, stage, and histology [2]. Melanoma accounts for 3% of all neoplasms in dogs and up to 7% of all malignant tumors, and it is the most common oral malignancy (56%) [3]. The majority of oral melanomas in dogs exhibit malignant behavior, characterized by rapid invasion of surrounding normal tissues, metastasis to the regional lymph nodes and lungs, and occasionally metastasis to other distant organs [4]. In dogs with melanoma, especially in the oral cavity, the primary tumor size and distant metastasis and/or lymph node metastasis have been considered to be prognostic factors [5]. Chemotherapy has some effect in controlling distant metastasis of the tumor and may help improve the outcome after other initial treatments, such as surgery. Therefore, the most effective treatment should include chemotherapy in the adjuvant setting after surgery [6].

The PI3K/AKT/mTOR signaling pathway is a central pathway that promotes cell growth, cell motility, protein synthesis, survival, and metabolism in response to hormones, growth factors, and nutrients [7]. Recent research results have shown that, in human malignant melanoma, the RAF–MEK–ERK (MAPK) and PI3K–AKT–mTOR (AKT) signaling pathways are constitutively activated via multiple mechanisms [8,9]. The MAPK and AKT signal transduction pathways regulate cell survival, proliferation, and invasion, which are key functions in the progression of melanoma [7]. Downstream S6K1 is activated along with the PI3K/AKT/mTORC1 pathway, which in turn initiates a series of cellular responses related to tumor formation. Among these pathways are vascular proliferation and tumor-cell proliferation [10]. Thus, there is an interest in developing new therapeutic approaches that inhibit the mammalian target of rapamycin (mTOR) activation for treating tumors.

The small molecule para-toluenesulfonamide (PTS) has been shown in vitro and in vivo to display activity against human hepatocellular carcinoma, human nonsmall-cell lung cancer, and human tongue squamous cell carcinoma [11,12,13]. PTS is a therapeutic drug that is injected directly into the tumors and has been shown to induce cancer cell death by activating apoptosis and necrosis in a variety of cancers, with minimal damage to normal tissues [11,13]. Moreover, it has good lipophilicity, which enables it to easily enter tumors and become well distributed around them. Its mechanisms of action include damage to lysosomes by inducing the permeabilization of their membranes. This causes the release of cathepsin B and induces lysosome-mediated cell death [13]. Furthermore, PTS inhibited cell proliferation in human castration-resistant prostate cancer (CRPC) cell lines via the arrest of the cell cycle during the G1 phase, independent of p21 and p27 (a process that included the downregulation of cyclin D1 and the inhibition of the phosphorylation of the retinoblastoma protein) [14]. It also induced apoptosis by causing a substantial loss of membrane potential in the mitochondria. This loss of membrane potential likely resulted from the upregulation of the apoptosis regulators BAX and p53, which belong to the proapoptotic Bcl-2 family. This small molecule was also able to inhibit the phosphorylation of mTOR, 4EBP1, and p70S6K in CRPC cell lines. The inhibition of mTOR/p70S6K signaling in PC-3 cells was abrogated by overexpressing constitutively active Akt, revealing that this pathway is AKT-dependent. Overall, these findings (both in vitro and in vivo) suggest that PTS is an effective antitumor agent that acts by inhibiting the mTOR/p70S6K pathways, which are both AKT-dependent and AKT-independent [14].

Cisplatin is a broad-spectrum anticancer chemotherapeutic agent that is used in canine oral cancer treatment and has antitumor activity against oral malignant melanoma and squamous cell carcinomas in dogs in combination with the anti-inflammatory agent piroxicam or radiation therapy [1,15,16,17]. The efficacy of cisplatin depends on its ability to induce DNA damage because cisplatin is activated upon its entry into the cytoplasm by displacing chloride atoms with water molecules, thus forming an electrophile with an affinity toward sulfhydryl groups on proteins and nitrogen donor atoms on nucleic acids [18]. Moreover, cisplatin, similar to many other chemotherapeutic drugs, can induce apoptosis; if the cells cannot repair the damage, they proceed to die [18,19,20]. The cellular toxicity of cisplatin results primarily from its ability to form intra- or interstrand crosslinks. The cells can become resistant through an enhanced ability to remove DNA adducts [21]. Cisplatin-based chemotherapy frequently results in acquired resistance via insufficient DNA binding, increased detoxification, increased DNA repair, deregulated expression of transporters, and altered expression and activation of genes involved in cell-death pathways, such as *p53*, *Bcl-2*, and *AKT/mTOR* [22]. Among these mechanisms, the activation of the AKT/mTOR pathway plays an important role in cisplatin resistance [23]. Boria et al. [1] treated oral melanomas with intravenous cisplatin (50–55 mg/m^2^) at 3-week intervals and with daily piroxicam (0.3 mg/kg), orally administered, and recorded an overall response rate of 18% and a median survival period of 119 days. In turn, Ahn et al. [24] reported that combination therapy of interferon beta and low-dose cisplatin significantly reduced canine melanoma tumor volume in a xenograft mouse model of canine melanoma. However, cisplatin resistance represents a problem for treatment, and other drugs are therefore often combined clinically with its application.

Here, we investigated the antitumor effects of the mTOR inhibitor PTS on melanoma cells. Furthermore, we examined its synergistic effects in combination with cisplatin. We implanted M5 canine melanoma cells in BALB/c nude mice and then assessed how tumor growth and aspects of cancer development (apoptosis, inflammation, and metastasis) were affected by these treatments.

## 2. Results

### 2.1. Tumor Growth Is Restricted by PTS

In the current study, treatment with both drugs (whether used individually or combined) led to smaller tumor size and volume relative to the control group (Figure 1a). Moreover, the time-course analysis showed that tumor growth was significantly retarded by all three treatments (*p* < 0.001) relative to the control (Figure 1b). These effects were strongest when PTS and cisplatin were used together; cisplatin alone reduced tumor mass by 48.7%, PTS alone reduced it by 44.2%, and their combination reduced it by 72.8% relative to the control (Table 1). Thus, the growth of canine melanoma xenotransplants was extremely slowed in a mouse model after treatment with both PTS and cisplatin.

### 2.2. The Abundance of TUNEL-Positive Cells Increases in Response to PTS

An essential aspect of the growth and survival of all organisms is apoptosis, that is, programmed cell death. In the later stages of apoptosis, the DNA of dying cells becomes substantially degraded, and cells with this degraded DNA can be detected using a TUNEL assay [25]. Our TUNEL assay revealed that all three treatments resulted in an abundance of apoptotic cells. In the cisplatin group, the TUNEL-positive cell count was 3.9 times that of the control (*p* < 0.01); in the PTS group, this ratio was 3.5 (*p* < 0.01); and in the cisplatin + PTS group, it was 8.6 (*p* < 0.001; Figure 2). The difference between the PTS and cisplatin groups was not significant, whereas that between the combined-treatment group and the other two treatment groups was significant (*p* < 0.001). Thus, all three treatments enhanced apoptosis, but the combined treatment was by far the most effective of the three approaches in this respect.

### 2.3. The Expression of Apoptosis-Associated Proteins Is Affected by PTS Treatment

During the orchestrated process of apoptosis, specific enzymes are activated that dissolve the nuclear components of the apoptotic cells, including the protein components of both the nucleus and the cytoplasm [26]. To further assess the effects of our treatments on apoptosis, we analyzed several proteins with related functions: the antiapoptosis protein B-cell lymphoma 2 (Bcl-2), extracellular signal-regulated kinase (ERK), phosphorylated ERK, and cleaved caspase 3 (Figure 3a). According to our Western blotting analysis, cleaved caspase 3 and phosphorylated ERK were significantly upregulated in the cisplatin and PTS (*p* < 0.01 for both) and cisplatin + PTS (*p* < 0.001 for both) groups compared with the control group (Figure 3b,c). The combination-treatment group also exhibited the upregulation of these proteins versus both of the individual-treatment groups (*p* < 0.01 for all except phosphorylated ERK in the cisplatin group, for which *p* < 0.05). The expression of Bcl-2, which is thought to suppress apoptosis, was lower in the individual-treatment groups (*p* < 0.05) and the combination-treatment group (*p* < 0.01) than in the control group (Figure 3d). There was no significant difference in Bcl-2 expression between the PTS and cisplatin groups, whereas its expression in the cisplatin + PTS group was significantly lower than that detected in both individual-treatment groups (*p* < 0.05 vs. PTS, *p* < 0.01 vs. cisplatin). Thus, both the PTS and the cisplatin + PTS treatments enhanced the expression of proteins that are associated with the promotion of apoptosis and reduced the expression of a protein that inhibits apoptosis, although the combination treatment had the strongest effect.

### 2.4. The Production of Cytokines Is Restricted by PTS Administration

There are strong links between inflammation and cancer, and combating inflammation may improve the efficacy of cancer prevention and therapies [27]. One such approach consists in the suppression of the expression of inflammatory cytokines. We conducted an immunohistochemical (IHC) analysis of the histopathology of the tumors from our mice, focusing on the inflammatory cytokines of interleukin-1β (IL-1β) (Figure 4a) and tumor necrosis factor-α (TNF-α) (Figure 4b). The activity of both molecules was reduced in the cisplatin group by 29.7% and 77.4%, respectively; in the PTS group, it was reduced by 26.5% and 73.0%, respectively; and in the cisplatin + PTS group, it was reduced by 75.0% and 92.2%, respectively, relative to the control group (Figure 4c,d). There was no significant difference between the PTS and cisplatin groups regarding either cytokine, and the effect was once again strongest in the combination-treatment group. The reduction in the cisplatin + PTS group relative to the PTS group was 65.9% for IL-1β and 71.3% for TNF-α, whereas relative to the cisplatin group, it was 64.4% for IL-1β and 65.8% for TNF-α (*p* < 0.01 for all). This demonstrates that the combination treatment was able to strongly suppress the expression of these inflammatory cytokines in canine melanoma.

We also conducted an enzyme-linked immunosorbent assay to assess the serum expression of interleukin-6 (IL-6), which is another inflammatory cytokine. We observed a pattern similar to that of IL-1β and TNF-α. Relative to the control group, the expression of IL-6 in the cisplatin, PTS, and cisplatin + PTS groups was reduced by 62.8%, 57.5%, and 76.8%, respectively (Figure 4e). Here again, the difference between the cisplatin and PTS groups was not significant, but the levels in the combination-treatment group were significantly lower than those in both individual-treatment groups (by 37.6% and 45.4%, respectively; *p* < 0.01 for both). Overall, these findings show that the administration of a combination of PTS and cisplatin strongly inhibited the expression of the inflammatory cytokines IL-1β, TNF-α, and IL-6 in canine melanoma.

### 2.5. The Expression of Inflammation-Related Factors Is Reduced in Response to PTS Administration

We conducted a Western blot analysis of factors related to inflammation. The expression of cyclooxygenase-2 (COX-2) in the cisplatin and PTS groups (*p* < 0.01) and the combination-treatment group (*p* < 0.001) was lower than that detected in the control group (Figure 5a). As described above, the difference between the cisplatin and PTS groups was not significant, whereas that between the combination-treatment group and the two individual-treatment groups was significant (*p* < 0.05; Figure 5b).

When cells are stimulated with inflammatory cytokines, IκB kinase (IκBα) degrades them, leading to the inhibition of nuclear factor-κB (NF-κB), which then accumulates and regulates the expression of specific genes [28]. In our analysis of *NF-κB* and *IκBα* mRNA expression, we found that the expression of *NF-κB* was significantly lower in all three experimental groups (*p* < 0.001 for all) than in the control group (Figure 5c). As expected, the reverse was true for *IκBα* expression (*p* < 0.001 for all three experimental groups vs. control; Figure 5d). For both genes, there was no significant difference between the two individual-treatment groups, whereas the differences between each of them and the combination-treatment group were significant (*p* < 0.01; Figure 5c,d). Overall, these findings indicate that PTS inhibited the expression of inflammation-related factors and responses, although this effect was not different from that of cisplatin. However, when the two agents were combined, the inhibition of the animals’ inflammatory response was stronger.

### 2.6. Factors Related to Metastasis Are Reduced by PTS Administration

Metastasis is one of the primary reasons for the high morbidity and mortality associated with cancer and is thought to explain approximately 90% of cancer deaths [29]. Therefore, we analyzed the expression of transforming growth factor β (TGF-β), CD44, epidermal growth factor receptor (EGFR), and vascular endothelial growth factor (VEGF), all of which are related to metastasis. We analyzed TGF-β and CD44 expression via IHC (Figure 6a,c) and that of EGFR and VEGF via Western blotting (Figure 6e). The expression of both TGF-β and CD44 was reduced in the cisplatin and PTS groups (*p* < 0.01 for all) and in the cisplatin + PTS group (*p* < 0.001 for both) relative to the control group (Figure 6b,d). TGF-β expression in the cisplatin, PTS, and cisplatin + PTS groups was reduced by 26.6%, 23.3%, and 62.6%, respectively, relative to the control group. Similarly, CD44 expression was reduced by 40.9%, 34.4%, and 61.6%, respectively. As reported above, the difference between the cisplatin and PTS groups was not significant, but significant differences were present between the combination-treatment group and the two individual-treatment groups (*p* < 0.01 for all); expression in the cisplatin + PTS group was reduced by 49.0% and 51.2% (TGF-β) and 35.1% and 41.5% (CD44) relative to the cisplatin and PTS groups, respectively.

The patterns of VEGF and EGFR expression were similar (Figure 6e). The expression of both of these factors was significantly lower in the individual-treatment groups (*p* < 0.05 for all) and the combination-treatment group (*p* < 0.001 for both) than in the control group (Figure 6f,g). Again, although there were no significant differences between the PTS and cisplatin groups, EGFR and VEGF expression in the cisplatin + PTS group was significantly lower than that detected in the remaining two treatment groups (*p* < 0.01 for all). This shows that PTS treatment was able to inhibit the expression of these metastasis-related factors and that combining it with cisplatin augmented this effect.

## 3. Discussion

Here, we investigated the effects of treatment with PTS alone and in combination with cisplatin on M5 canine melanoma tumors implanted in nude mice. These mice have been used as an effective model for assessing the efficacy of treatments on ectopically xenotransplanted human carcinoma cells [30,31]. Tumor growth may be particularly favorable in such mice because they lack a thymus and, thus, cannot produce T cells [32]. We found that, similar to cisplatin, PTS treatment reduced tumor size and weight. Even though the effects of these two agents were similarly suppressive of tumors in all of our analyses, they achieved a substantially stronger antitumor effect when they were combined. Our findings indicated that this combination treatment promoted apoptosis by upregulating cleaved caspase 3 and phosphorylated ERK and downregulating Bcl-2. Furthermore, it reduced inflammation by restricting the production of IL-1β, TNF-α, IL-6, COX-2, and *NF-κB*. Finally, it rendered metastasis less likely by suppressing TGF-β, CD44, VEGF, and EGFR.

The mTOR pathway is critical for regulating the cell cycle; it is an important downstream signaling pathway that is activated by many biological functions and plays a major role in regulating autophagy [33,34]. When this pathway is aberrantly activated, it produces signals that both promote tumor-cell growth and metastasis and enable these cells to invade healthy tissues [35]. Many clinical and histopathological similarities between human and canine melanoma were recognized by the Comparative Melanoma Tumor Board consensus study of the human relevance of melanoma in pet dogs [36]. Moreover, the study indicated that AKT and mTOR, as well as their downstream product, p70S6K, are present and active in canine melanoma cells. The activation of the mTOR pathway can be inhibited by rapamycin; in turn, the treatment of melanoma cells with rapamycin decreases the surviving tumor-cell fraction [37]. We were able to demonstrate that the inhibition of the mTOR pathway is an effective approach for treating cancer, and the research interest in inhibitors of this pathway has increased as a result. In the present study, we found that the proliferation of M5 canine melanoma was restricted by PTS, suggesting that this approach may be useful in a clinical setting.

In human cancers, chemotherapy using a combination of an anticancer drug and a cell-signal inhibitor has been found in several studies to yield a better response than either drug in isolation. For example, combining rapamycin with sorafenib (a multikinase inhibitor that inhibits Raf-1 and B-RAF) synergistically reduces melanoma cell proliferation, and targeting these two types of signaling pathways simultaneously, may be more effective for treating melanoma than using either agent alone [38]. Research into drug combinations in canine melanoma treatment has also been conducted. The combination of the MEK inhibitor trametinib and the dual PI3K/mTOR inhibitor dactolisib synergistically decreased cell survival in association with caspase 3/7 activation and altered the expression of cell-cycle regulatory proteins and Bcl-2-family proteins [39]. Therefore, we assessed the inhibitory and synergistic effects of combining cisplatin with PTS to treat mice implanted with M5 canine melanoma cells. We found that, although PTS alone exhibited inhibitory effects, its combination with cisplatin strongly and synergistically inhibited tumor-cell growth. Alternatively, because of the small sample size of the mouse model, particularly serum volume, the toxic effects of PTS combined with cisplatin treatment were not estimated in this study. Previous reports indicated that the liver dysfunction induced by the intraperitoneal injection of a single dose of cisplatin (7.5 mg/kg body weight) on day 5 of the experiment was characterized by elevated serum levels of transaminases (aspartate aminotransferase (AST) and alanine transaminase (ALT)) [40,41]. In contrast, the serum ALT and AST levels were unaffected in rats treated with a single dose of 2 mg/kg of cisplatin [42]. Moreover, 4 years after the last PTS treatment, patients with hepatocellular carcinoma treated with PTS exhibited no abnormal liver function serum values [11]. To ensure that such liver damage is identified in a timely manner, clinicians should closely monitor the hepatic function of dogs with melanoma receiving PTS combined with cisplatin.

We then examined factors related to apoptosis. This is an important aspect of tumors because cancer cells are able to evade apoptosis despite having abnormalities and replicate themselves continuously. Chemotherapeutic approaches force the tumor cells to undergo apoptosis by causing cellular distress or DNA damage, in turn triggering cell-death signals [43]. We conducted TUNEL and DAPI assays to investigate the extent to which apoptosis occurred in the tumor cells in the mice. Similar to that observed for tumor growth, PTS on its own was beneficial as it enhanced tumor-cell apoptosis. However, in combination with cisplatin, the treatment strongly promoted apoptosis. Therefore, the latter treatment reduced the ability of tumor cells to evade apoptosis, thereby limiting tumor growth. We also observed the upregulation of cleaved caspase 3 and phosphorylated ERK, both of which are key for apoptosis, in mice treated with the combination regimen. Antiapoptotic Bcl-2 contributes to cancer formation and progression by promoting the survival of altered cells [44]. A previous study showed that an siRNA directed against the canine *Bcl-2* gene reduced Bcl-2 mRNA and protein expression in a canine malignant oral melanoma cell line (MCM-N1) and resulted in both a decrease in the number of viable cells and an increase in the apoptotic cell rate [45]. These findings indicate the important role of Bcl-2 activity in the inhibition of apoptosis in canine melanoma cells and reinforce the notion of Bcl-2 as a putative therapeutic target in tumors. We also found that PTS triggered the downregulation of *S6K1* and *mTOR* (Appendix A). As mentioned above, PTS must be able to suppress the expression of mTOR. In contrast, because the mTOR signaling pathway regulates autophagy and tumor-cell apoptosis, it can also promote tumor occurrence and progression. Sun et al. concluded that mTOR inhibits the expression of glycogen synthase kinase-3 (GSK-3) in prostate cancer cells. The downregulation of GSK-3, in turn, inhibits the caspase-3 signaling pathway, leading to the reduction in ROS production. Decreased ROS inhibit apoptosis in tumor cells to protect prostate cancer cells [46]. Accordingly, we found that PTS alone and combined with cisplatin triggered a significant upregulation of cleaved caspase 3 and phosphorylated ERK relative to the control group. This was probably caused by the inhibition of the mTOR pathway. Interestingly, in addition to its ability to induce DNA damage, cisplatin induces tumor-cell apoptosis [18,19]. PTS combined with cisplatin enhanced the melanoma apoptosis afforded by the cisplatin alone treatment, as assessed by the IHC staining and Western blot analyses of apoptosis-related proteins performed in this study. Overall, these findings indicated that maximal antitumor effects were obtained when this PTS therapy was combined with a DNA-damaging chemotherapeutic agent. This study demonstrated the possible applicability of PTS for treating canine melanoma.

There is a well-established close link between chronic inflammation and tumor formation. Therefore, inflammation is thought to be a key characteristic of cancer [47]. In fact, various solid tumors maintain an inflammatory immune microenvironment, which enhances tumor development and metastasis [48]. Thus, we investigated the effects of PTS treatment on various factors related to inflammation (IL-1β, TNF-α, IL-6, COX-2, *NF-κB*, and *IκBα*). The expression levels of the *IL-1β* gene or IL-1β protein are associated with the invasiveness and metastasis of melanoma [49]. Metastatic melanoma cell lines do not secrete IL-1β; rather, they promote IL-1β production from macrophages in mice [50]. Moreover, a link between high levels of TNF-α and an increased risk of tumor formation and development has been described in vivo [51]. The effects of COX-2 in melanomas are thought to be largely caused by its role in the production of prostaglandins, especially prostaglandin E_2_ [52]. Furthermore, in melanoma cells overexpressing COX-2, an increase in prostaglandin E_2_ levels and expression of prostaglandin E_2_ receptors resulted in the promotion of cell migration [53]. The transcription factor NF-κB regulates inflammatory responses by enhancing the expression of specific cellular genes, which is further linked to the promotion of carcinogenesis [54]. COX-2 is a major molecular target of NF-κB. Various inflammatory stimuli and mediators have been demonstrated to increase COX-2 expression via the activation of NF-κB, thus eliciting inflammation and consequent tumorigenesis [55,56]. In addition, IL-6, as one of the targets of NF-κB, can be regulated by STAT3 activation [57]. IL-6 was expressed in both human and dog melanomas. A previous study demonstrated that the prosurvival function of NF-κB was related to its functional interaction with the PI3K/AKT/mTOR signaling pathway [58]. With respect to factors that inhibit inflammation, the stimulation of TNF-α activates IκB, which is an important kinase that acts downstream in the TNF-α signaling pathway. It then phosphorylates IκBα, which causes its degradation by the ubiquitin–proteasome proteolytic system [59]. The susceptibility of malignant cells to apoptosis-inducing factors is increased by inhibiting NF-κB [60]. Therefore, NF-κB and IκBα play important roles in a range of inflammatory responses and immune processes. Here, the administration of PTS reduced the expression of IL-1β, TNF-α, IL-6, COX-2, *NF-κB*, and *IκBα*. Furthermore, C-reactive protein (CRP) is synthesized mainly as a result of stimulation by proinflammatory cytokines, and higher lung cancer risk and tumor progression are associated with elevated CRP levels [61]. Thus, it makes sense that the level of CRP was lowest in mice treated with both PTS and cisplatin (Appendix A).

The combination treatment also combated inflammation by suppressing inflammatory cytokines and promoting anti-inflammation mediators. This is another way in which PTS contributes to retarding tumor growth. In addition, the nuclear protein Ki67 is a marker of proliferation at inflamed sites and is strongly associated with tumor growth [62], to the extent that, in canine melanoma, the relative abundance of Ki67-positive tumor cells is used as a prognostic factor [63]. In fact, the Ki67 index differs significantly between malignant and benign melanocytic neoplasms in dogs and correlates negatively with survival [64]. We found that PTS, both alone and in combination with cisplatin, suppressed Ki67 expression (Appendix A). Overall, these findings showed that PTS effectively inhibited tumor-growth and antiapoptotic factors in canine melanoma cells by suppressing inflammation and tumor progression.

Metastasis is another key factor that determines cancer prognosis. The proliferation and migration of heavily transformed tumor cells are stimulated by TGF-β, resulting in metastasis and tumor progression [65]. In canine oncology, little information is available regarding the association between anticancer immunity and the TGF-β pathway. A previous study found that the mean plasma TGF-β1 levels in tumor-bearing dogs were significantly higher than those in healthy controls [66]. Moreover, CD44 is a cell-surface molecule that mediates cell adhesion and communication with the extracellular matrix. In the case of melanoma, CD44 has been implicated in cell migration and proliferation in vitro [67,68]. CD44 is recognized as a cancer stem cell marker in canine breast tumors, and its expression is also increased in canine leukemia, melanoma, and osteosarcoma [69,70,71,72]. We found that PTS, used alone or in combination with cisplatin, substantially suppressed TGF-β and CD44 expression, thereby reducing the metastatic potential of the tumor cells. It is also likely that the suppression of TGF-β expression enhances the efficacy of drug therapy by helping to inhibit metastasis [73]. Furthermore, VEGF strongly promotes angiogenesis, and its overexpression is linked to tumor progression and metastasis [74]. Angiogenesis represents a fundamental step in the malignant growth of tumors and metastasis. Although many pro- and antiangiogenic factors affect the formation of new blood vessels, the central growth factor in this process is VEGF. Furthermore, VEGF expression appears to be related to tumor grade and prognosis in some malignancies [75]. Overexpression of VEGF in human melanoma results in a phenotype that has increased malignant potential compared with melanomas with low VEGF expression [76]. EGFR also has potential as an important therapeutic target in human cancer because it may be involved in the progression of cutaneous melanomas specifically [77] and more generally acts upstream of mTOR as a signal transducer of mitogens, which play a role in cancer pathogenesis and development [78]. In addition, the expression of the zinc finger E-box-binding homeobox 1 (ZEB1), a transcription factor that accelerates migration and invasion, indicates epithelial–mesenchymal transition (EMT) in canine melanoma cells [79]. Moreover, EMT-inducing ZEB1 promotes cancer cell metastasis and loss of cell polarity [80]. The aberrant expression of ZEB1 has been reported in a variety of human cancers, where it is generally believed to foster migration, invasion, and metastasis via mTOR signaling [81,82]. We found that treatment with PTS, both alone and in combination with cisplatin, reduced the expression of the mTOR/S6K1 signaling pathway, resulting in the suppression of the expression of the *ZEB1* mRNA (Appendix A), possibly leading to a reduction in the potential for the M5 canine melanoma cells to metastasize. Therefore, our findings suggest that PTS exerts antitumor effects through the suppression of VEGF and EGFR expression, especially when administered in combination with cisplatin.

The tumors of canine and human melanomas are generally resistant to chemotherapy and radiation therapy. In veterinary oncology, no standard of care for melanoma has been firmly established. The treatment of dogs with melanoma consists primarily of surgery, with the options of hypofractionated or definitive radiation therapy and platinum chemotherapy [1,83,84,85]. Multiple veterinary studies have investigated therapeutic approaches incorporating carboplatin, cisplatin, and melphalan but only achieved poor response rates and no improvements in survival time. This is particularly the case when chemotherapeutic drugs are tested as the sole treatment or as adjuvant therapy after surgery or radiation therapy [1,86,87,88]. The development and metastasis of tumors are closely related to the tumor microenvironment, which varies in structure and function [89]. The tumor microenvironment can affect tumor progression, prognosis, and the efficacy of immunotherapy, with tumor-infiltrating lymphocytes being reportedly central to these effects [90]. We performed IHC staining using CD45, a marker of these lymphocytes, and found that CD45 activity was 1.4, 1.3, and 1.8 times higher in the cisplatin, PTS, and cisplatin + PTS groups, respectively, than in the control group (Appendix A). This further confirmed that PTS plays a key role in inhibiting these canine melanomas, although this effect was strongest in combination with cisplatin. This combined therapy significantly inhibited the growth of M5 cells by promoting apoptosis, suppressing inflammation, and reducing the potential for metastasis.

Melanomas are malignant neoplasms originating from melanocytes. Canine melanoma xenograft mouse models have been valuable in elucidating the mechanisms of malignant transformation, disease progression, and drug resistance in cutaneous melanoma [91,92]. However, mouse models for noncutaneous melanomas are still lacking, and mice have constraints, such as limited population heterogeneity, tightly controlled environmental living conditions, and the difficulty of obtaining serial tissue samples [93]. Accordingly, although melanomas occur in most animal species, the dog is considered the best animal model for this disease [16]. Alternatively, the route of PTS administration via local intratumoral injection can reveal the therapeutic effects of canine melanoma in xenografted mice. Moreover, the intratumoral injection of PTS is a palliative treatment that aims to further improve the survival and quality of life of patients with advanced or recurrent carcinomas, patients with cancer with severe comorbidities, or patients with a poor performance status [94]. However, a few published studies have shown that the association between chemotherapy with other therapy routes may potentially slow local progression and/or improve overall survival in canine melanoma [16,95]. Thus, future studies should focus on the individualized antitumor effect of therapies, representing a potential and powerful strategy for the treatment of melanomas in dogs, such as the combination of PTS with radiation therapy or PTS with chemotherapy.

## 4. Materials and Methods

### 4.1. Animals and Cell Line

We obtained germ-free male 6-week-old BALB/cByJNarl mice for use in this study (National Laboratory Animal Breeding and Research Center, Taipei, Taiwan). We housed two mice in each cage and provided them with sterile food and water. The mice had ad libitum access to Laboratory Rodent Diet 5058 (Lab Diet; Purina Mills, St. Louis, MO, USA) and were allowed to acclimate for 1 week prior to the collection of baseline weekly body-weight and food-intake changes. The mice were maintained in microisolation cages on HEPA-filtered ventilated racks (Rungshin IVC Systems, Taichung, Taiwan) under germ-free conditions with constant temperature (22 ± 2 °C) and relative humidity (55 ± 5%) and a 12:12 h light:dark cycle. The bedding of the cages consisted of autoclaved shredded wood chips (Tapvei Oy, Kaavi, Finland). The cages were equipped with a variety of sterile toys of different shapes: tunnels, tubes, and triangles (Young Li, Taipei, Taiwan). The Institutional Animal Care and Use Committee (IACUC) at the National Chiayi University (IACUC Approval No. 110003) reviewed and approved our study protocol; the procedures of which were in accordance with the Guidelines for the Care and Use of Laboratory Animals published by the Taiwanese Ministry of Health and Welfare.

We obtained the M5 canine melanoma cell line from the School of Veterinary Medicine at National Taiwan University [96]. We cultured the cells in a humidified atmosphere (95% air, 5% CO_2_) at 37 °C in 90% high-glucose Dulbecco’s Modified Eagle’s Medium supplemented with 5% fetal bovine serum, 50 IU/mL of penicillin, and 50 mg/mL of streptomycin (Gibco Laboratories, Grand Island, NY, USA). We routinely passaged the cells by removing the medium and overlaying the cell monolayer with 0.25% trypsin and 0.1% ethylenediaminetetraacetic acid.

### 4.2. Inoculation and Treatment of Tumors

The experimental treatments were commenced after the implanted M5 cells had formed a detectable tumor mass, which was assessed on day 7 after implantation. Briefly, we anesthetized the mice via intraperitoneal injection with Zoletil 50 (10 mg/kg; Virbac Taiwan, Taipei, Taiwan) and then subcutaneously injected 100 µL of a cell suspension containing 10^7^ viable M5 cells into a posterior leg. We assessed the development of the tumor lesions on day 7 and randomly divided the animals that showed a distinct tumor of 4–5 mm in diameter [97] into four groups (seven animals per group). The control group received saline, and the three treatment groups received their treatments three times per week. The cisplatin group received 2 mg/kg cisplatin (Sigma, St. Louis, MO, USA) via intraperitoneal injection. The PTS group was administered 100 mg/kg PTS (Gongwin Biopharm Holding, Taipei, Taiwan) through local intratumoral injection. Finally, the cisplatin + PTS group received 100 mg/kg PTS and 2 mg/kg cisplatin. The dosage of cisplatin used here was based on studies investigating how this drug affects apoptosis, invasion, metastasis, angiogenesis, and the growth signal mechanisms in canine melanomas implanted in mice [24]. We based the PTS dosage on the dose that is generally used to treat lung cancer [12], and the route of PTS administration was according to that reported in various studies [12,13,94,98]. We assessed the growth of the tumors every 7 days by measuring their largest and smallest diameters and calculated their volume according to the following formula: *V* = 0.5 × *a* × *b*^2^, where *a* and *b* are the largest and smallest diameters, respectively.

### 4.3. Clinical Observations and Histopathological Analysis

We observed the mice daily for clinical signs, weighed them every 3 days, and, at 35 days after tumor-cell implantation, euthanized them with an overdose of anesthetic (1.2 mg/kg urethane, intraperitoneally; Sigma) combined with carbon dioxide. Immediately before sacrificing the mice, we collected blood under anesthesia for hematological assessment. After euthanizing the mice, we immediately excised the tumors. We divided the tumor specimens into two groups. We treated one group with 10% formalin and embedded the specimens in paraffin. We investigated these specimens by subjecting them to hematoxylin and eosin staining; terminal deoxyribonucleotidyl transferase (TdT)-mediated biotin-16-dUTP nick-end labeling (TUNEL assay, APO-BrdU TUNEL Assay Kit; BD Pharmingen, San Diego, CA, USA); and IHC for IL-1β, TNF-α, TGF-β, and CD44, using primary antibodies against these factors (Merck, Billerica, MA, USA). We assessed protein expression via IHC with the TAlink mouse/rabbit polymer detection system produced by BioTnA (Kaohsiung, Taiwan). We used a high-resolution digital microscope (Moticam 2300; Motic Instruments, Richmond, BC, Canada) equipped with Motic Images Plus (version 2.0) to capture and analyze the images. Regarding the other group of tumor specimens, we first minced them coarsely, homogenized them, and then stored them in a freezer at −80 °C. We later subjected them to Western blotting to analyze the expression of cleaved caspase 3, ERK, Bcl-2, COX-2, VEGF, and EGFR.

### 4.4. Measurement of the Serum Levels of IL-6

We measured the serum levels of IL-6 in the blood samples using commercial mouse kits (ab213749 and ab100697; Abcam, Cambridge, MA, USA) according to the manufacturer’s protocol. We added 50 μL of samples to each of the wells in 96-well antibody-coated plates and then incubated these for 2 h at room temperature. We then loaded 50 μL of the detector antibody solution into each well and incubated the plates for an additional 1 h at room temperature. Next, we added 50 μL of HRP–streptavidin solution (ab210901; Abcam) to each well and once again incubated the plates for 1 h at room temperature. Finally, we added 100 μL of tetramethylbenzidine substrate to each well and incubated the plates for another 10 min in the dark at room temperature. We stopped the reaction by adding 100 μL of stop solution. We read the absorbance at a wavelength of 450 nm and expressed the results in pg/mL.

### 4.5. RNA Extraction and Real-Time Quantitative Polymerase Chain Reaction

We used the TRI Reagent (Sigma) to extract the total RNA from the tumor tissues. We quantified the RNA concentrations based on absorbance at 260–280 and 230–260 nm using a Qubit fluorometer (Invitrogen, Carlsbad, CA, USA). We reverse transcribed 1 μg of the RNA into cDNA using an iScript cDNA synthesis kit (Bio-Rad, Hercules, CA, USA) according to the manufacturer’s instructions. We performed a real-time polymerase chain reaction (PCR) using the cDNA and iTaq universal SYBR Green supermix (Bio-Rad) according to the manufacturer’s protocol. We quantified the mRNA expression levels of *NF-κB* and *IκBα* using the CFX Connect Real-Time PCR Detection System (Bio-Rad). We used the following settings for PCR: 40 cycles of 95 °C for 30 s; 95 °C for 15 s, 60 °C for 30 s; and a final 5 min at 72 °C. We used the following sequence primers for *NF-κB*, *IκBα*, and *β-actin*: *NF-κB*: forward, 5′-ATGGCTTCTATGAGGCTGAG-3′ and reverse, 5′-GTTGTTGTTGGTCTGGATGC-3′; *IκBα*: forward, 5′-GCCCTTGTCCCTGTCCCTA-3′ and reverse, 5′-GCAGAGTATTTCCCTTTGGTTTGA-3′; and *β-actin*: forward, 5′-ACTGGAACGGTGAAGGTGACA-3′ and reverse, 5′-ATGGCAAGGGACTTCCTGTAAC-3′ [99,100]. We calculated the expression levels of the two target genes relative to the *β-actin* levels and expressed them using the 2^−ΔΔCt^ method.

### 4.6. Western Blotting

We performed Western blotting on the second group of tumor specimens according to protocols reported in previous studies [101,102]. We used antibodies against β-actin, Bcl-2, COX-2, VEGF, and EGFR (Sigma) and against cleaved caspase 3, phosphorylated ERK (threonine 202/tyrosine 204), and ERK (Cell Signaling Technology, Beverly, MA, USA). We employed enhanced chemiluminescence reagents (Thermo Scientific, Rockford, MA, USA) to assess immunoreactivity and UVP ChemStudio (Analytik Jena, Upland, CA, USA) to detect signals. We quantified protein expression and phosphorylation using ImageJ (National Institutes of Health, Bethesda, MA, USA).

### 4.7. Statistical Analysis

We presented all of the data as mean ± standard deviation. We used *t*-tests to test for significant differences between pairs of groups. When comparing more than two groups, we performed an analysis of variance with a post-hoc Bonferroni correction. *p*-values below 0.05, 0.01, and 0.001 were considered significant, very significant, and extremely significant, respectively.

## 5. Conclusions

In conclusion, this study highlighted the co-effect of PTS and cisplatin for the potential treatment of canine melanoma. Our findings evidenced that PTS alone reduces the growth rate of canine melanoma xenotransplants in mice and that combining PTS with a traditional chemotherapy drug such as cisplatin yields a stronger inhibitory effect. This observation was associated with the inhibition of tumor growth via an apoptotic mechanism in canine melanoma cells, where the apoptotic expression of cleaved caspase 3 and ERK phosphorylation was enhanced, and the antiapoptotic expression of Bcl-2 was reduced. Furthermore, we found that PTS, both alone and combined with cisplatin, suppressed inflammatory cytokine (IL-1β, TNF-α, and IL-6) production, which may have retarded tumorigenesis. Treatments that have antitumor and anti-inflammatory effects result in lower levels of COX-2 and the *NF-κB* mRNA and higher levels of the *IκBα* mRNA. The inhibitory effect of PTS on the in-tumor mTOR pathway resulted in the downregulation of TGF-β, CD44, VEGF, and EGFR, which would have had the effect of retarding tumor-cell migration and invasion, thus reducing the potential risk of metastasis. Overall, our in vivo findings regarding these xenograft-induced tumors in nude mice showed that PTS had a range of pharmacological effects in this experiment; it promoted apoptosis, suppressed inflammation, and supported mechanisms that suppress tumor metastasis. Moreover, combining cisplatin with PTS resulted in a stronger effect than using PTS alone. These findings suggest an avenue for investigating and developing additional therapeutic approaches for the treatment of canine melanoma specifically and cancers in general.

## Figures and Tables

**Figure 1 animals-12-02272-f001:**
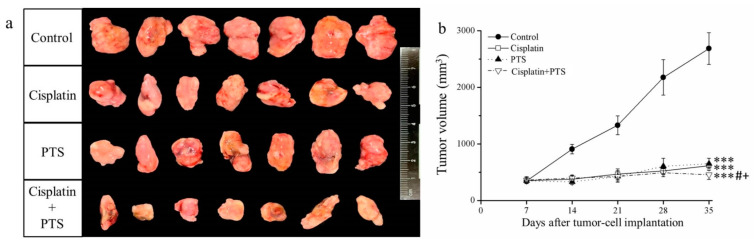
Effects of treatment with cisplatin, para-toluenesulfonamide (PTS), and their combination on M5 canine melanoma tumor growth in BALB/c nude mice. (**a**) Tumors excised from the mice 35 days after tumor-cell implantation. (**b**) Change in tumor volume over time. Treatments were administered from day 7 (when the tumors were detected) three times per week, as follows: saline (control), 2 mg/kg cisplatin (cisplatin), 100 mg/kg PTS (PTS), and 100 mg/kg PTS combined with 2 mg/kg cisplatin (cisplatin + PTS). Data are presented as mean ± standard deviation (SD), *n* = 7 per group. *** *p* < 0.001 vs. control; ^#^
*p* < 0.05 vs. cisplatin; ^+^
*p* < 0.05 vs. PTS.

**Figure 2 animals-12-02272-f002:**
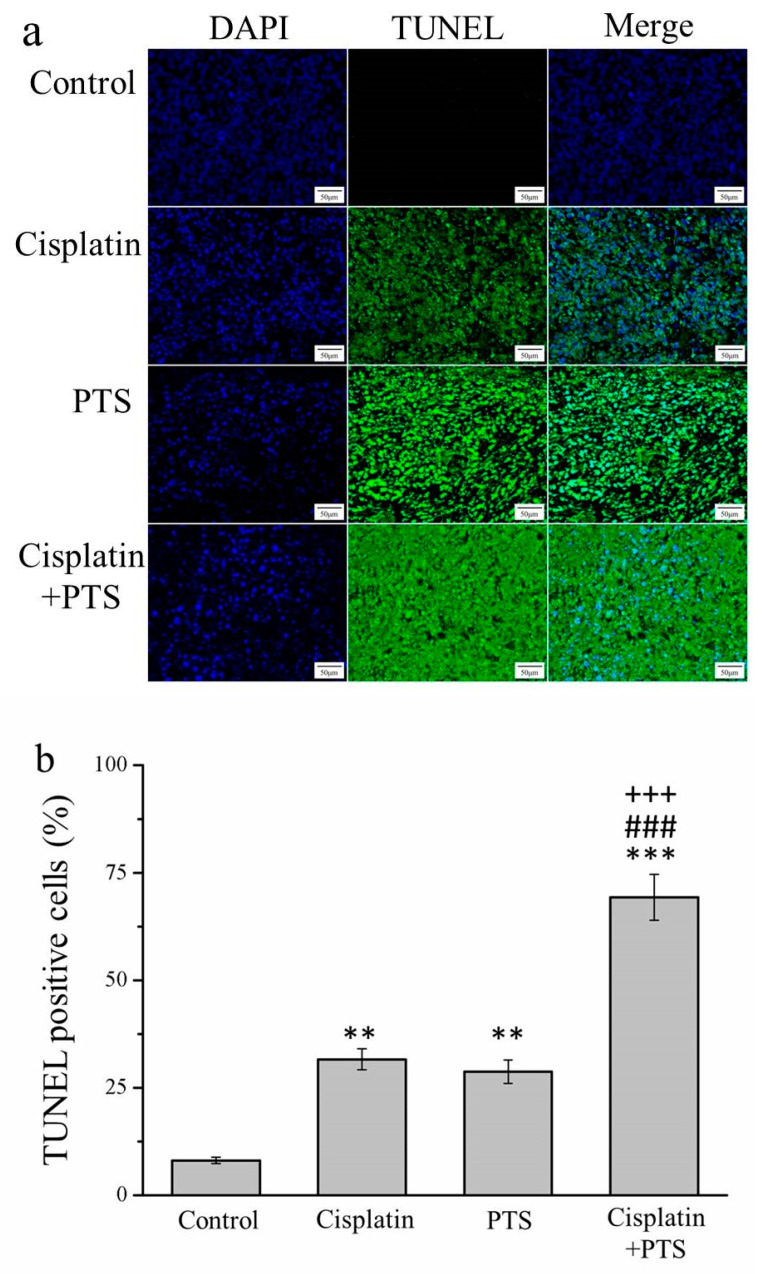
TUNEL/DAPI assay of apoptosis in M5 canine melanoma tumors implanted in BALB/c nude mice. (**a**) Staining with DAPI, TUNEL, and their merging. (**b**) Percentage of cells in the tumors that were TUNEL-positive in control mice and those administered with one of the following treatments three times per week, starting on day 7 (when the tumors were detected): saline (control), 2 mg/kg cisplatin (cisplatin), 100 mg/kg PTS (PTS), or 100 mg/kg PTS and 2 mg/kg cisplatin (cisplatin + PTS). Data are presented as mean ± SD, *n* = 7 per group. ** *p* < 0.01, *** *p* < 0.001 vs. control; ^###^
*p* < 0.001 vs. cisplatin; ^+++^
*p* < 0.001 vs. PTS. Scale bars: 50 µm; magnification: 200×.

**Figure 3 animals-12-02272-f003:**
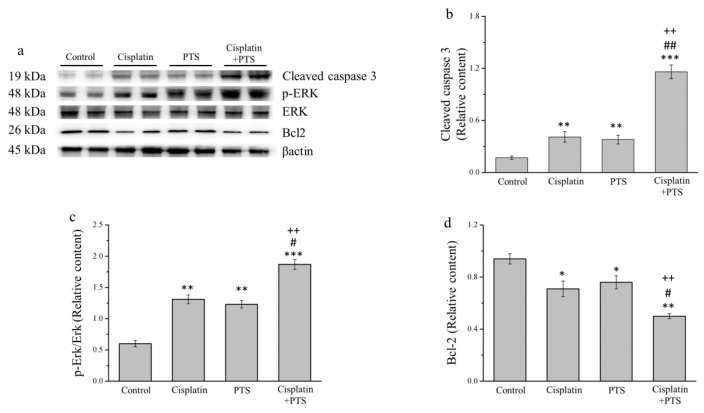
Western blot analysis of apoptosis-related proteins in M5 canine melanoma tumors implanted in BALB/c nude mice. (**a**) Representative Western blot. (**b**–**d**) Quantitative comparisons of the expression of cleaved caspase 3 (**b**), phosphorylated ERK (**c**), and (**d**) Bcl-2. The mice were administered the following treatments three times per week: saline (control), 2 mg/kg cisplatin (cisplatin), 100 mg/kg PTS (PTS), or 100 mg/kg PTS and 2 mg/kg cisplatin (cisplatin + PTS). Data are presented as mean ± SD, *n* = 7 per group. * *p* < 0.05, ** *p* < 0.01, and *** *p* < 0.001 vs. control; ^#^
*p* < 0.05 and ^##^
*p* < 0.01 vs. cisplatin; ^++^
*p* < 0.01 vs. PTS.

**Figure 4 animals-12-02272-f004:**
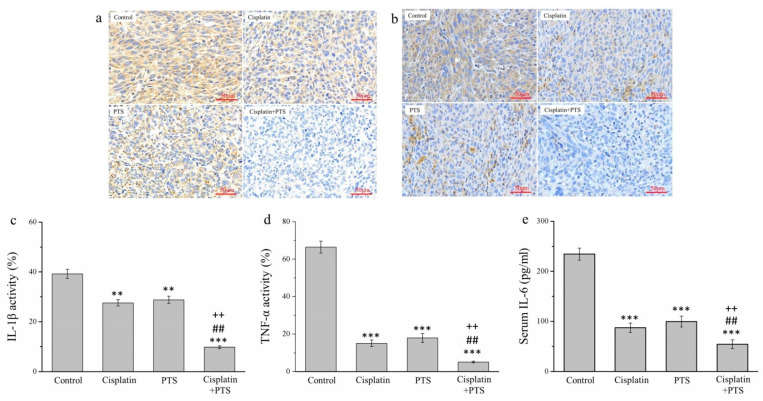
Immunohistochemical (IHC) analysis of the expression of three inflammatory cytokines in M5 canine melanoma tumors implanted in BALB/c nude mice. (**a**,**b**) Representative images showing IL-1β (**a**) and TNF-α (**b**) expression. (**c**–**e**) Quantitative analyses of the IHC expression of IL-1β (**c**), TNF-α (**d**), and serum IL-6 (**e**). The mice were administered the following treatments three times per week: saline (control), 2 mg/kg cisplatin (cisplatin), 100 mg/kg PTS (PTS), or 100 mg/kg PTS and 2 mg/kg cisplatin (cisplatin + PTS). Data are presented as mean ± SD, *n* = 7 per group. ** *p* < 0.01 and *** *p* < 0.001 vs. control; ^##^
*p* < 0.01 vs. cisplatin; ^++^
*p* < 0.01 vs. PTS. Scale bars: 50 µm.

**Figure 5 animals-12-02272-f005:**
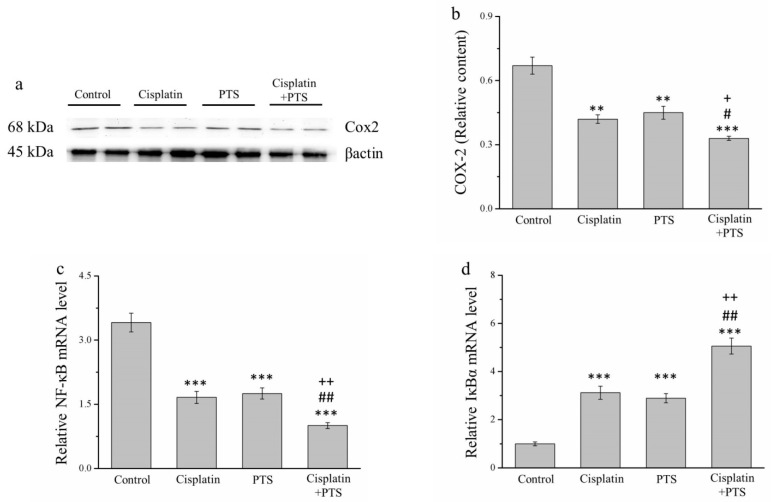
Analysis of factors related to inflammation in M5 canine melanoma tumors implanted in BALB/c nude mice. (**a**) Representative Western blot of cyclooxygenase-2 (COX-2). (**b**) Quantitative comparison of COX-2 expression in the four treatment groups. We conducted a quantitative polymerase chain reaction (PCR) to assess the relative mRNA expression levels of (**c**) *NF-κB* and (**d**) *IκBα*. The mice were administered the following treatments three times per week: saline (control), 2 mg/kg cisplatin (cisplatin), 100 mg/kg PTS (PTS), or 100 mg/kg PTS and 2 mg/kg cisplatin (cisplatin + PTS). Data are presented as mean ± SD, *n* = 7 per group. ** *p* < 0.01 and *** *p* < 0.001 vs. control; ^#^
*p* < 0.05 and ^##^
*p* < 0.01 vs. cisplatin; ^+^
*p* < 0.05 and ^++^
*p* < 0.01 vs. PTS.

**Figure 6 animals-12-02272-f006:**
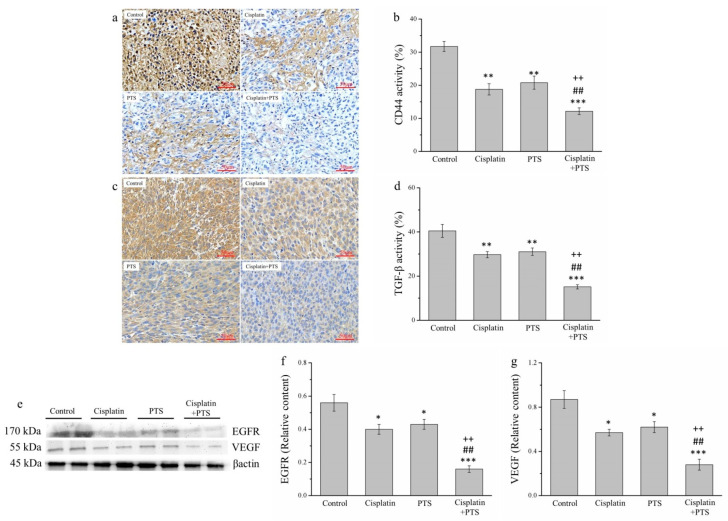
Analysis of factors related to metastasis in M5 canine melanoma tumors implanted in BALB/c nude mice. (**a**–**d**) Representative images (**a**,**c**) and quantitative comparisons (**b**,**d**) from our IHC analysis of CD44 (**a**,**b**) and TGF-β (**c**,**d**) expression. (**e**–**g**) Representative Western blot (**e**) from our analysis of the relative expression of EGFR (**f**) and VEGF (**g**). The mice were administered the following treatments three times per week: saline (control), 2 mg/kg cisplatin (cisplatin), 100 mg/kg PTS (PTS), or 100 mg/kg PTS and 2 mg/kg cisplatin (cisplatin + PTS). Data are presented as mean ± SD, *n* = 7 per group. * *p* < 0.05, ** *p* < 0.01, and *** *p* < 0.001 vs. control; ^##^
*p* < 0.01 vs. cisplatin; ^++^
*p* < 0.01 vs. PTS. Scale bars: 50 µm.

**Table 1 animals-12-02272-t001:** Tumor masses after excision from sacrificed BALB/cByJNarl mice.

	Tumor Mass (g)	Tumor Mass Reduction (%)
Control	2.67 ± 0.32	–
Cisplatin	1.37 ± 0.25 ***	48.7
PTS	1.48 ± 0.33 ***	44.2
PTS + cisplatin	0.72 ± 0.19 ***^#+^	72.8

Data are presented as mean ± SD, *n* = 7. *** *p* < 0.001 vs. control; ^#^
*p* < 0.05 vs. cisplatin; ^+^
*p* < 0.05 vs. PTS.

## Data Availability

The data presented in this study are available on request from the corresponding author.

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
