# Peer review of "Effects of Para-Toluenesulfonamide on Canine Melanoma Xenotransplants in a BALB/c Nude Mouse Model"

_animals, 2022, doi:10.3390/ani12172272_

Round 1
Reviewer 1 Report
Review remarks:
suggest shortening and changing title, perhaps like that (but it is only a suggestion):
"Effects of para-toluenesulfonamide on canine melanoma xenotransplants in a BALB/c nude mouse model"
Line 60: melanoma in the dog is not really a “commonly occurring tumor”; (below you correctly say 7% of….)
Line 61: avoid specifying or listing locations other than oral mucosa and eye; melanoma can appear as primary tumor nearly everywhere; the point is that in locations other than oral mucosa or eye, such as skin (with exception of phalanges), melanoma has more benign characteristics.
Line 70:…has some effect….
Line 76: please specify that these studies were done on human melanoma (if so) since you are here reporting on canine melanoma cells
Line 80: can you call them “physiological” if they lead to tumor formation ?
Line 82: instead of “need” say perhaps “interest”
Line 86: specify in which species; human ?
Line 95: in the dog ?
were observers blinded to the drug and/or placebo applications ?
Line 104: This referenced (15) statement is misleading: as it stands, it insinuates that cisplatin is (clinically) effective against malignant melanoma in the dog; in fact, in the referenced study cisplatin has not been tested clinically but on tissues in vitro. Similarly, reference 19 evaluates cisplatin effect on canine melanoma xenografts in a mouse model; the cited reference no 1 (Boria et al.) is the only clinical trial of cisplatin in canine melanoma showing an only moderate effect (oral melanoma regression in 2 out of 11 dogs). Thus, cisplatin is not commonly used to treat melanoma in dogs, which contrasts with human medicine where cisplatin seems frequently used in melanoma chemotherapy protocols. Please clarify within your text, where appropriate, that 1) no extrapolations to clinical chemotherapy protocols in dogs with malignant melanoma should be made and 2) that the experimental design and primary goal of your present study is meant (I believe) to improve treatment protocols for human melanoma (using a canine melanoma cell transplant model). Also, when citing references please make clear whether referring to the canine or human species and whether studies were done in vitro or in vivo (that is in patients).
Line 127: this was already explained in the Introduction
Line 134: “Canine melanoma growth is thus strongly inhibited by treatment with both PTS and cisplatin.” This sentence is misleading: in fact, what you are showing is that the growth of canine melanoma xenotransplants is slowed in a mouse model –
Table 1: cells were implanted on day 0, and the first visible ? palpable ? tumor was detected on day 7; so, in the graphic, why do on day 0 the implantation sites show a tumor volume of ~ 300 cubic mm ? (it seems that 0 point on the x axis in the graph is in fact day 7 of the experiment; please clarify or adjust graph.
Line 149: delete “greater”
Line 165: nature is neither “careful” nor “careless”; please re-phrase
Discussion:
Materials & Methods: just a remark to style: it is preferrable to describe in the 3rd person and in the past tense; that is ”samples were taken….” And not: ‘we took samples, we did… “
Conclusions:
Line 545: again, results of this study on mice is just an indication that the combination of these compounds may have some effecti in (clinical) canine melanoma; thus the word “potential” should change place in the first sentence of Conclusions:
“In conclusion, this study highlights the co-effect of PTS and cisplatin for the potential treatment of canine melanoma.”
Line 546:…reduces the growth rate of canine melanoma xenotransplants in mice….
Line 557: ….reducing the potential risk of metastasis.” (in your study you did not evaluate metastasis, so this is a hypothetical conclusion).
Line 563: ….melanoma in specific, and …..
Reviewer 2 Report
This paper (animals-1795076) is a significant study aiming to create a highly effective treatment for melanoma, for which no effective treatment has been established in dogs, by combining PTS and cisplatin, a novel anti-tumor compound in the veterinary field. the combination of PTS and cisplatin has shown a high apoptosis induction effect, anti-inflammatory effect and the combination of PTS and cisplatin showed high apoptosis-inducing, anti-inflammatory, and metastasis-inhibiting effects, which is hopeful for the treatment of canine melanoma.
 However, I feel that an objective evaluation of the results presented in this paper is lacking, and we suggest the following improvements.
1. Throughout the paper, there seems to be a lack of objective statements as to why the combination of cisplatin and PTS is synergistic compared to each as monotherapy. In particular, there is a lack of description of the relationship between the inhibitory effect of cisplatin on tumor cell DNA synthesis and the effect of PTS. A description of the mechanism of action of cisplatin should be added to the “introduction”, and the rationale for the synergistic effect of combination therapy should be added to the “discussion” with reference to the mechanism of action of cisplatin.
2. I recognize that immunosuppression with cisplatin is a side effect and a bad symptom, but can combination therapy with PTS, which enhances it, be evaluated as a good phenomenon? This should be added to the discussion.
3. Lane 356: Change PST to PTS
4. Lane 419: The author states that the combination of cisplatin and PTS suppressed ZEB1 mRNA expression, which resulted in metastasis inhibition, but the prediction of the mechanism that led to this phenomenon is not described and should be added.
Reviewer 3 Report
The article provides new insights about a possible drug combination between cisplatin and PTS for melanoma treatment in dogs using a immunodeficient mouse model. The aim of the article is clear; however, some improvement could be done, especially by stating the limitations of the study and by clarifying some design choices. More information regarding the methods should also be provided. The following should be addressed:
1. Revise English and rephrase some sentences that are less clear such as the one in line 32-33.
2. No information is provided regarding the acclimatization time, type of bedding, type of cage, microbiological characterization of the animals and environmental enrichment used.
3. No information was provided also regarding the type of food and water provided for the animals.
4. Where treatments randomised by cage?
5. Which humane endpoints were used for this experiment?
6. The route of administration of the cells is not clear: was it subcutaneous administration or intramuscular? This should be stated in a more clear way and a justification for this location should also be provided as more commonly this xenografts are performed at the cervical dorsal area.
7. Were other tissued collected to investigate potential metastasis. This route of cells administration is uncommon to cause metastasis so other models of induction could be used if metastasis were of interest. This should at least be referred at the discussion, as well as other limitations of the study such as the fact that the treatment route (IP) is only applicable to rodents but not to other species of interest.
8. Did the authors explore possible toxic effects of this combination? Some sentence regarding the effects that were observed on the general status of the animals should be provided, and, if toxicity was not evaluated on key tissues, this limitation should also be addressed.
9. There is some misspelling between PTS and PST.
10. The treatment frequency for the control group was daily and for the other groups was 3 times a week according to figure 1 information. Please explain why the experiment was designed in such a way.
11. Treatments were started when tumours were 4-5 mm. Why did authors choose only one measure to define the tumor size at the treatment initiation and not a volume approximation?
12. The dose of zoletil used is missing, as well as the intraperitoneal treatment administration volumes.
13. The euthanasia methods is also not clear: which anaesthetic was used and dose.
